# Wastewater Treatment and Electricity Production in a Microbial Fuel Cell with Cu–B Alloy as the Cathode Catalyst

**Paweł P. Włodarczyk *** and **Barbara Włodarczyk**

Institute of Technical Sciences, Faculty of Natural Sciences and Technology, University of Opole, Dmowskiego Str. 7-9, 45-365 Opole, Poland
* Correspondence: pawel.wlodarczyk@uni.opole.pl; Tel.: +48-077-401-6717

**Abstract:** The possibility of wastewater treatment and electricity production using a microbial fuel cell with Cu–B alloy as the cathode catalyst is presented in this paper. Our research covered the catalyst preparation; measurements of the electroless potential of electrodes with the Cu–B catalyst, measurements of the influence of anodic charge on the catalytic activity of the Cu–B alloy, electricity production in a microbial fuel cell (with a Cu–B cathode), and a comparison of changes in the concentration of chemical oxygen demand (COD), $NH_4^+$, and $NO_3^-$ in three reactors: one excluding aeration, one with aeration, and during microbial fuel cell operation (with a Cu–B cathode). During the experiments, electricity production equal to 0.21–0.35 mA·cm$^{-2}$ was obtained. The use of a microbial fuel cell (MFC) with Cu–B offers a similar reduction time for COD to that resulting from the application of aeration. The measured reduction of $NH_4^+$ was unchanged when compared with cases employing MFCs, and it was found that effectiveness of about 90% can be achieved for $NO_3^-$ reduction. From the results of this study, we conclude that Cu–B can be employed to play the role of a cathode catalyst in applications of microbial fuel cells employed for wastewater treatment and the production of electricity.

**Keywords:** non-precious metal catalysts; Cu–B alloy; microbial fuel cell; cathode; environmental engineering; oxygen electrode; renewable energy sources

## 1. Introduction

At present, the power industry faces difficulties ensuring the production of greater volumes of energy to meet the increased demand. Simultaneously, the production of waste and wastewater increases considerably. This means that large amounts of industrial and municipal wastewater may be generated. The traditional design of a wastewater treatment plant consumes a lot of energy to perform efficiently, and this generates considerable costs. Approximately $23 billion is spent annually by the United States on domestic wastewater treatment and improving the quality of publicly owned treatment infrastructure costs another $200 billion [1]. In this context, it is clear that it is important to decrease the costs of wastewater treatment. Nowadays, there are different ideas for the use of wastewater as a raw material for other technologies, and there has been fast development in renewable sources of energy using wastewater. A technical device that can combine electricity production with wastewater treatment is a microbial fuel cell (MFC) [2]. MFCs are ecological sources of electric energy which produce electricity from wastewater [2–4]. While the first observation of an electrical current generated by bacteria is generally credited to Potter [5], very few practical advances were achieved in this field prior to the 1960s [6–8]. In the 1990s, there was increased interest in MFC research [9–11], but significant development of MFCs only occurred in recent years [2–4,12–15].

MFCs are bio-electrochemical systems in the form of devices that use bacteria as catalysts to oxidize organic and inorganic matter and generate a current [2,7]. Activated sludge is capable of producing electrons e$^-$ and H$^+$ ions. In an MFC, organic material is oxidized on the anode, and the product of oxidation is $CO_2$ and electrons. For a glucose reaction, we obtain [2,16,17].

$$\text{ANODE } C_6H_{12}O_6 + 6H_2O \rightarrow 6CO_2 + 24H^+ + 24e^- \tag{1}$$

$$\text{CATHODE } 24H^+ + 24e^- + 6O_2 \rightarrow 12H_2O \tag{2}$$

$$\text{Summary reaction}: \ C_6H_{12}O_6 + 6O_2 \rightarrow 6CO_2 + 6H_2O + \text{electricity} \tag{3}$$

Electron-producing bacteria that are capable of wastewater treatment play a key role in the effective performance of MFCs [2,4]. Such bacteria include *Geobacter*, *Shewanella*, or *Pseudomonas*, among many other genera [18–25]. An analysis of reports in this field demonstrates that the highest values of capacity are generated by MFCs comprising multispecies aggregates, where microorganisms grow in the form of biofilms. Mixed cultures seem to provide a solid and more efficient solution compared to cultures based on a single strain, and their isolation from natural sources is a much less complex task. In contrast, the use of single-strain cultures is associated with technical limitations, mainly resulting from the need for ensuring sterile growth conditions, and the process usually involves high costs [26]. Figure 1 shows a diagram of the microbial fuel cell.

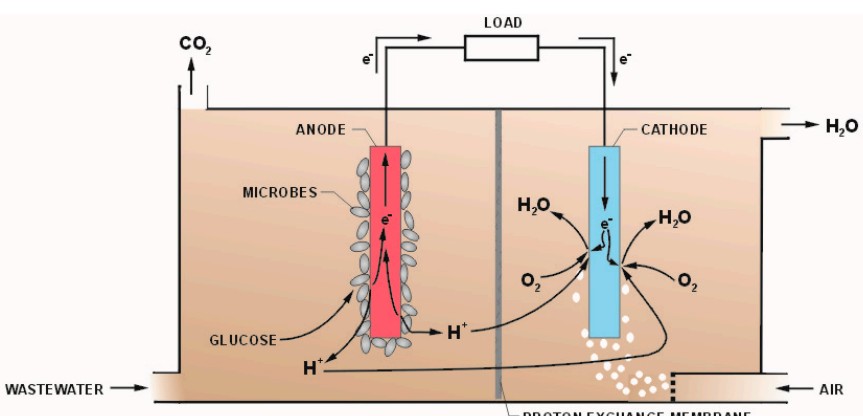

**Figure 1.** Operating principles of a microbial fuel cell (MFC). Figure is not to scale.

Currently, several theoretical and practical works connected to increasing the MFCs' power have been presented, not only in the field of the microorganism selection. The upper limit of the power level that is achievable in MFCs is not yet known because there are many reasons for power limitations. The reasons limiting the maximum power density may be different, e.g., high internal resistance or low speed of reactions on electrodes [2,4,27,28]. The speed of the process depends on the catalyst used. In an MFC, the catalyst at the anode is microbes (on a carbon electrode). Thus, it is important to find a catalyst for the cathode. Due to its excellent catalytic properties, platinum is most commonly used as the catalyst. However, due to the high price of platinum, we should look for other catalysts, such as the non-precious metals. In previous works [29–34] were compared the performances of microbial fuel cells (MFCs) equipped with different cheap electrode materials (graphite, carbon felt, foam, and cloth and carbon nanotube sponges, and Polypyrrole/carbon black composite) during two-month-long tests in which they were operated under the same operating conditions. Despite using sp2 carbon materials (carbon felt, foam, and cloth) as the anode in the different MFCs, the results demonstrated that there were important differences in the performance, pointing out the relevance of the surface area and other physical characteristics to the efficiency of MFCs. Differences were found not only in the production of electricity but also in the consumption of fuel. Carbon felt was found to be the most efficient anode material, whereas the worst results were obtained with carbon cloth. Performance seems to have a

direct relationship with the specific area of the anode materials. In comparing the performances of the MFCs equipped with carbon felt and stainless steel as the cathodes, the latter showed the worse performance, which clearly indicates how the cathodic process may become the bottleneck of the MFC performance. Besides platinum, graphite, carbon felt, foam, cloth, etc., Ni or metal borides are frequently used as the catalyst of electrodes. Due to costs, in MFCs, carbon or carbon cloth with platinum is most often used as the cathode catalyst. It is also possible to use metal catalysts for the cathodes of MFCs [35,36]. The theoretical current density is described by the Butler–Volmer exponential function [37]. Unfortunately, in real conditions, the choice of catalyst is mainly come to by experimental methods [37,38]. For this reason, experimental research on the selection of new catalysts for MFCs is still conducted [14,29,30,33–40]. Herein, we demonstrate the possibility of using Cu–B alloy as a cathode catalyst for MFCs for municipal wastewater treatment and electricity production.

## 2. Results and Discussion

Figures 2–5 show the trend in time of electroless potential measured at Cu–B alloys in alkaline electrolyte (KOH). Cu–B alloys that contained 3%, 6%, 9%, and 12% of B, oxidized for 1, 3, 6, and 8 h, were selected and the experimental set up in 3.2 chapter was adopted for these measurements.

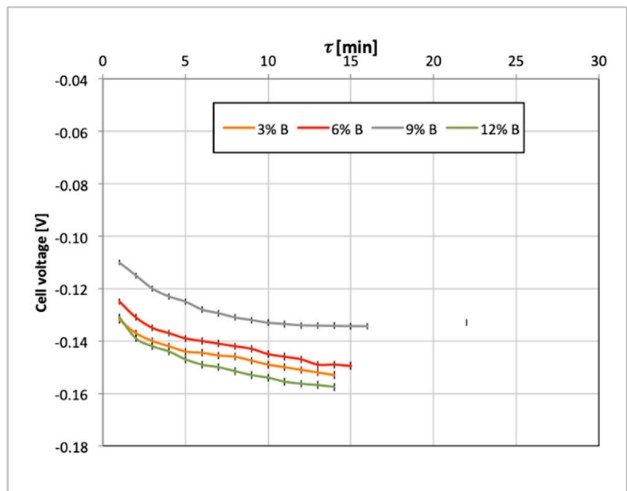

**Figure 2.** The electroless potential of electrodes with Cu–B catalyst which were oxidized for 1 h.

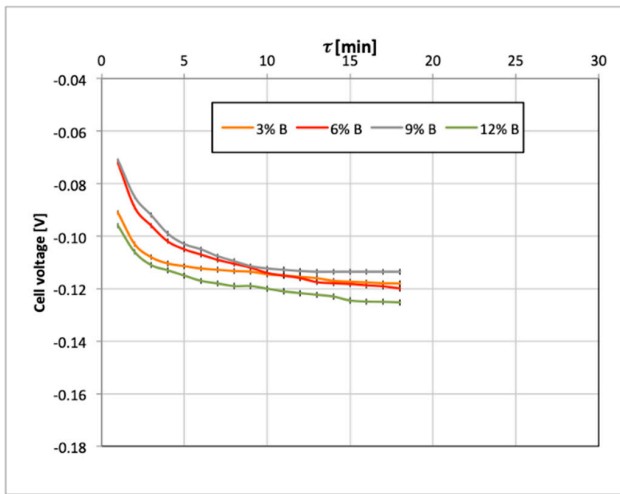

**Figure 3.** The electroless potential of electrodes with Cu–B catalyst which were oxidized for 3 h.

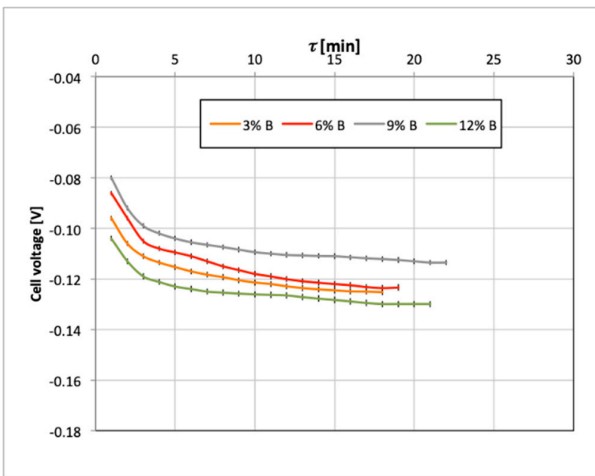

**Figure 4.** The electroless potential of electrodes with Cu–B catalyst which were oxidized for 6 h.

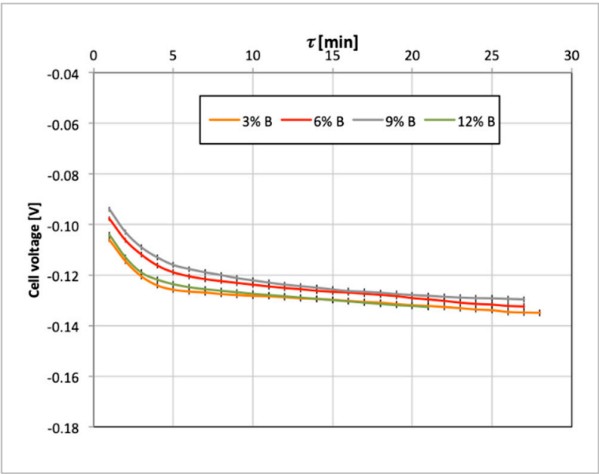

**Figure 5.** The electroless potential of electrodes with Cu–B catalyst which were oxidized for 8 h.

For all concentrations of boride in the Cu–B catalyst, higher current density was obtained when it was oxidized for 6 h. Thus, measurements of the effect of anodic charge on the catalytic activity of the Cu–B catalyst were performed for the samples oxidized for 6 h, shown in Figures 6–10. By colored lines (1-4) was marked the subsequent anodic charge.

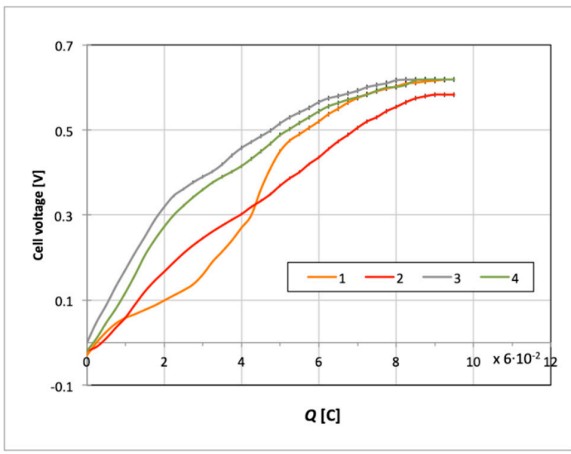

**Figure 6.** Influence of anodic charge on the catalytic activity of Cu–B alloy containing 3% of B.

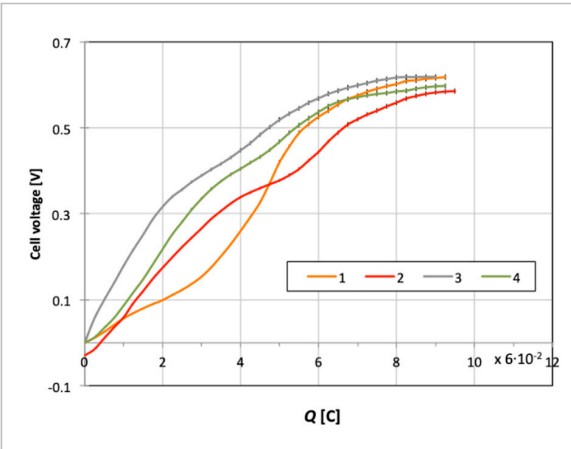

**Figure 7.** Influence of anodic charge on the catalytic activity of Cu–B alloy containing 6% of B.

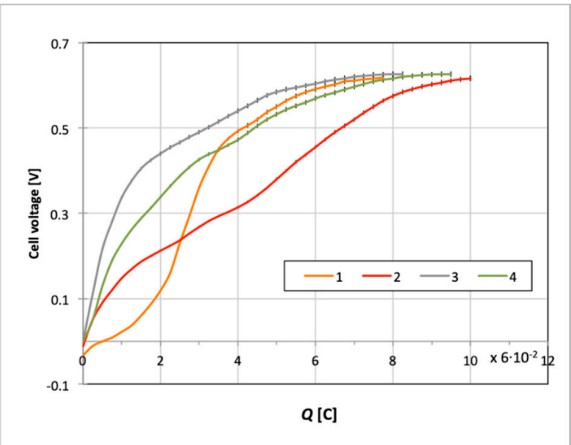

**Figure 8.** Influence of anodic charge on the catalytic activity of Cu–B alloy containing 9% of B.

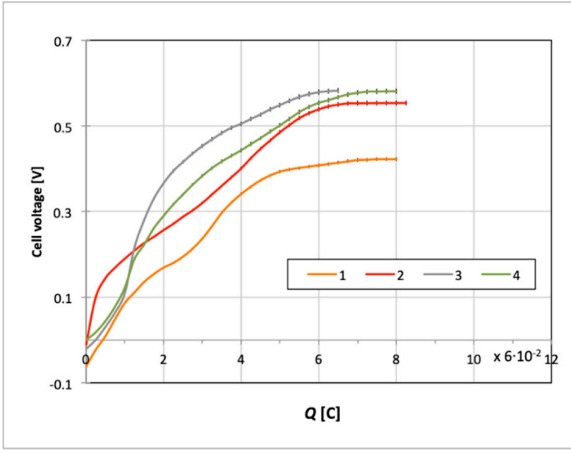

**Figure 9.** Influence of anodic charge on the catalytic activity of Cu–B alloy containing 12% of B.

Based on the data (Figures 2–5), it should be noted that in any case, the electroless potential is the highest for the alloy with 9% B concentration. Moreover, the cell voltage is also the highest for the alloy with 9% B concentration after the third anodic charging of the electrode (Figures 6–9). Such parameters ensure high efficiency of the electrode's functioning. Therefore, based on the data analysis (Figures 2–9), the electrode with 9% B concentration after the third anodic charge was chosen for further measurements of the MFC.

Figures 10–12 show the trends of chemical oxygen demand (COD), $NH_4^+$, and $NO_3^-$ concentration during wastewater treatment in the three reactors (R1-R3, 3.3 chapter), in which the MFC was equipped with a Cu–B cathode and with a carbon cloth cathode.

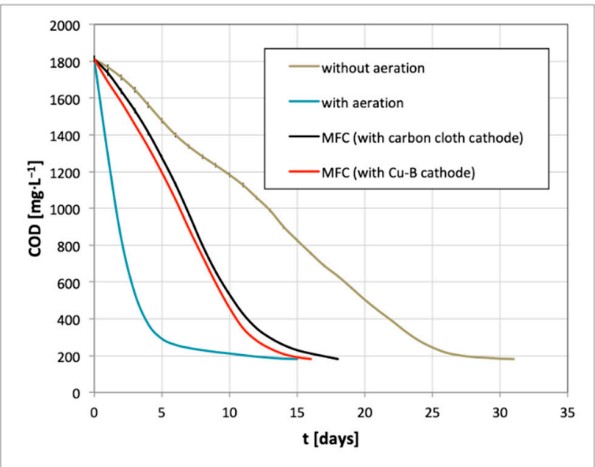

**Figure 10.** Trend of chemical oxygen demand (COD) reduction during wastewater treatment performed at the different reactors.

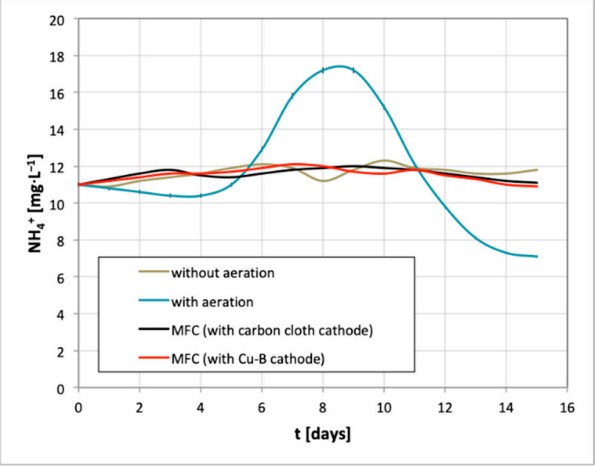

**Figure 11.** Trend of $NH_4^+$ reduction during wastewater treatment performed at the different reactors.

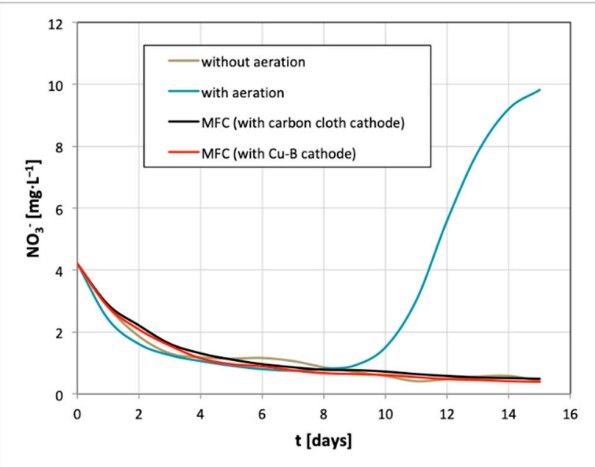

**Figure 12.** Trend of $NO_3^-$ reduction during wastewater treatment performed at the different reactors.

Figure 13 shows power curves of the MFC (with a Cu–B cathode and with a carbon cloth cathode). The data shown in Figure 13 were obtained during the operation of the MFC (R3 in 3.3 chapter).

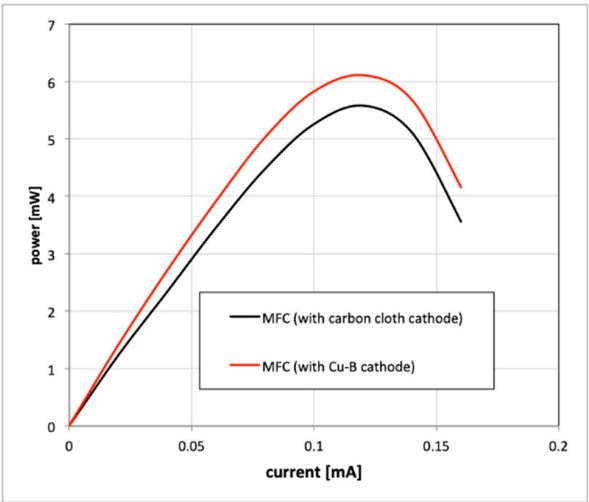

**Figure 13.** Power curves of the MFC: effect of the cathode material.

An analysis of the data indicated that, for all the examined concentrations of B, Cu–B alloy oxidized for 6 h showed high electroless potential. The best result was obtained after triple anodic oxidation, at a temperature of 673 K and when the concentration of boride is equal to 9%. Therefore, the electrode with a 9% concentration of B after the third anodic charge was chosen for further measurements with the MFC. Wastewater from a wastewater treatment plant was fed into the MFC. The R3 reactor was analyzed in two cases: using the Cu–B cathode and using a carbon cloth cathode. A removal level of COD of 90% was recorded in all reactors (R1, R2, and R3) (Figure 10). The characteristics of the curves were found to be different. Better performance was found in the characteristic curve of COD removal during aeration (Figure 10, blue line) than in the curve of COD removal during MFC operation (Figure 10, red and black line) since around 81% effectiveness of COD reduction after about 5 days was obtained in this period.

However, over time, a COD reduction level of 90% was achieved by the application of the MFC (over a period of 16 days resulting from the use of the Cu–B cathode, compared to 18 days for the case of the carbon cloth cathode). These results are similar to the case of the reduction time during the aeration process (15 days). It should be noted that the performance of the Cu–B cathode is better than that of carbon cloth. A faster decrease in the COD concentration is always measured at Cu-B sample. The measurement of $NH_4^+$ reduction shows no changes during the MFC operation (R3) (Figure 11) for either the MFC with a Cu–B cathode or that with a carbon cloth cathode. A similar situation occurred for the R1 without aeration.

However, the measurements (Figure 12) also show the effectiveness of $NO_3^-$ reduction (during 15 days) in the MFC with a Cu–B cathode (effectiveness of 90.71%), the MFC with a carbon cloth cathode (effectiveness of 88.33%), and the reactor without aeration (effectiveness of 89.52%). The increased $NH_4^+$ concentration in R2 results from the attachment of hydrogen molecules to ammonia ions during wastewater putrefaction (Figure 11) [41,42]. The increased $NO_3^-$ concentration (Figure 12) is the result of nitrification during the growth of bacteria [43]. During the MFC experiments, current densities of 0.21 mA·cm$^{-2}$ for the carbon cloth cathode and 0.35 mA·cm$^{-2}$ for the Cu–B cathode were obtained. Power levels of 5.58 mW in the MFC with the carbon cloth cathode and 6.11 mW in the MFC with the Cu–B cathode were obtained. The power obtained in the MFC with Cu–B alloy as the cathode catalyst is similar to the power obtained in an MFC with a Ni–Co cathode [44,45]. However, in the case of using the Ni–Co cathode, the MFC was powered with process wastewater from a yeast factory,

while the MFC (with Cu–B alloy as the cathode catalyst) analyzed in this work was powered with municipal wastewater from a wastewater treatment plant.

## 3. Materials and Methods

### 3.1. Preparation of a Cathode with Cu–B Catalyst

The Cu–B alloys were obtained by the method of electrochemical deposition and were deposited on copper mesh electrodes. The alloys were deposited from a mixture of mainly $NaBH_4$ and $CuSO_4$ [46,47]. The alloys were obtained at temperatures of 355–365 K and at a current density 1–3 A·dm$^{-2}$ [42,46,47]. The composition of the mixture used for electrochemical catalyst deposition is summarized in Table 1.

**Table 1.** Composition of the mixture applied for catalyst deposition (Cu–B alloy).

| Component | Volume |
| --- | --- |
| $NaBH_4$ | 0.02 mol·L$^{-1}$ |
| $CuSO_4 \cdot 7H_2O$ | 0.05 mol·L$^{-1}$ |
| NaOH | 1.00 mol·L$^{-1}$ |
| Trilon B | 0.12 mol·L$^{-1}$ |

Before the deposition of the alloy, the copper electrode was prepared in several steps [42,44,48,49]: the surface was mechanically purified (to a shine) and then degreased in 25% aqueous solution of KOH (after degreasing, the surface should be completely wettable with water); then, the electrode was digested in acetic acid and subsequently washed with alcohol.

To obtain different contents of B in the alloys, the temperature and current density were selected experimentally. Electrodes with Cu–B alloy as the catalyst (a selection of alloys with different contents of B) for further measurements were selected by the XRD method using a single-crystal X-ray diffractometer (Xcalibur, Oxford Diffraction, UK). During the electrochemical deposition, 12 alloys with different concentrations of boride were obtained. Figure 14 shows the concentrations of components in the samples obtained during electrochemical deposition.

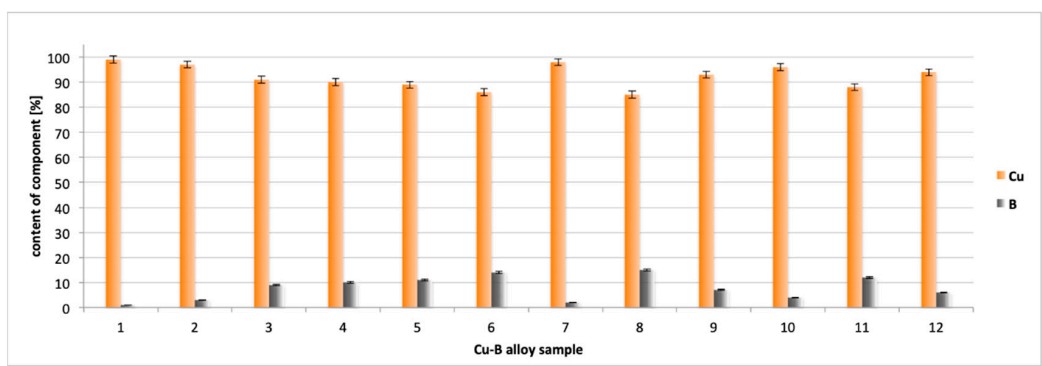

**Figure 14.** Concentration of components in the samples obtained during electrochemical deposition.

For further research, Samples 2 (3% of B), 3 (9% of B), 11 (12% of B), and 12 (6% of B) were selected. A further increase in B concentration (over 12%) did not cause an increase in efficiency of the MFC (i.e., an increase in cell power and current density). These samples were selected based on previous studies [42,46,47] and to ensure an even increase in B concentration to 12% (in this case, every 3%). Thus, the Cu–B alloys with 3%, 6%, 9%, and 12% of B were used in measurements.

### 3.2. Selection of the Electrodes (with Cu–B Catalyst) for Measurements

To assess the Cu–B alloy oxygen activity, first, the oxidation of the alloy was carried out with measurements of the stationary potential of the oxidized electrode. Due to the fact that the cathode is

constantly oxygenated during MFC operation, it is necessary to pre-oxidize it. Without pre-oxygenation, the electrode would oxidize during MFC operation and there would be an efficiency decrease (and, thus, also a decrease in the current density and the cell's power). The Cu–B alloy was oxidized at a temperature of 673 K. The oxidation times were 1, 3, 6, and 8 h. The KS 520/14 silt furnace (ELIOG Industrieofenbau GmbH, Römhild, Germany) was used for electrode oxidation. Next, we measured the influence of anodic charge on the catalytic activity of the Cu–B alloy. Initial anode charging avoids a drop in the cell (MFC) efficiency during operation. Figure 15 shows a schematic view of the measurement of the catalytic activity of the Cu–B alloy.

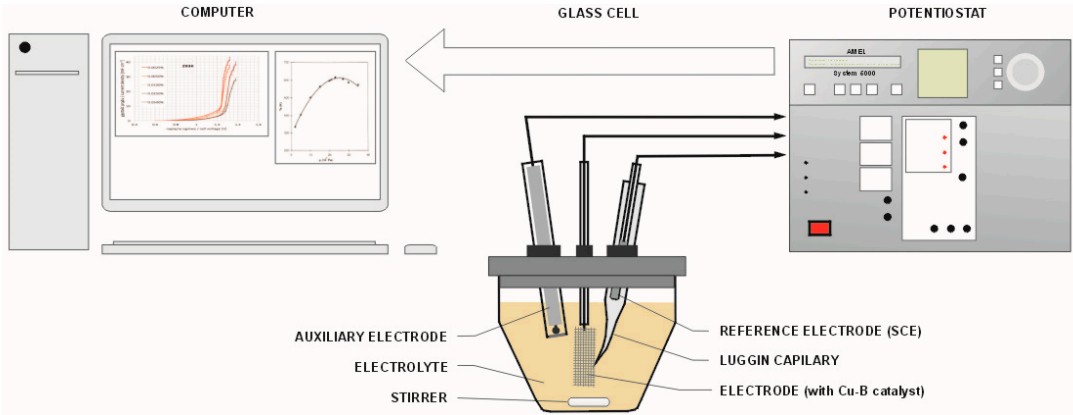

**Figure 15.** Schematic view of the reactor for the measurement of the electroless potential and the influence of anodic charge of electrodes with a Cu–B catalyst.

These measurements were carried out in a glass cell with the use of a potentiostat. An aqueous solution of KOH (2 M) was used as the electrolyte. A saturated calomel electrode (SCE) was used as the reference electrode. The experiments were conducted using an AMEL System 500 potentiostat (Amel S.l.r., Milano, Italy) with CorrWare software (Scribner Associates Inc., Southern Pines, NC, USA).

### 3.3. Measurements of Wastewater Treatment and Electricity Production in the MFC with a Cu–B Cathode

Wastewater samples from a municipal wastewater treatment plant were used in the measurements applied for the purposes of this study. Table 2 contains a summary of the parameters of the wastewater applied in the measurements.

**Table 2.** Parameters of wastewater applied for measurements.

| Parameter | Value |
|:---:|:---:|
| COD [$mg \cdot L^{-1}$] | 1811.0 |
| $NH_4^+$ [$mg \cdot L^{-1}$] | 11.1 |
| $NO_3^-$ [$mg \cdot L^{-1}$] | 4.2 |
| pH | 6.5 |

The initial phase of the analysis involved measuring the reduction in the chemical oxygen demand (COD) in the investigated samples. Subsequently, variations in the $NH_4^+$ and $NO_3^-$ concentrations were measured. These measurements were conducted with regard to three types of reactors: one excluding aeration (Reactor 1—R1), one with aeration (Reactor 2—R2), and in the form of a continuous measurement performed in an MFC (Reactor 3—R3). Figure 16 shows these three types of reactors used in the measurement of wastewater parameters during wastewater treatment.

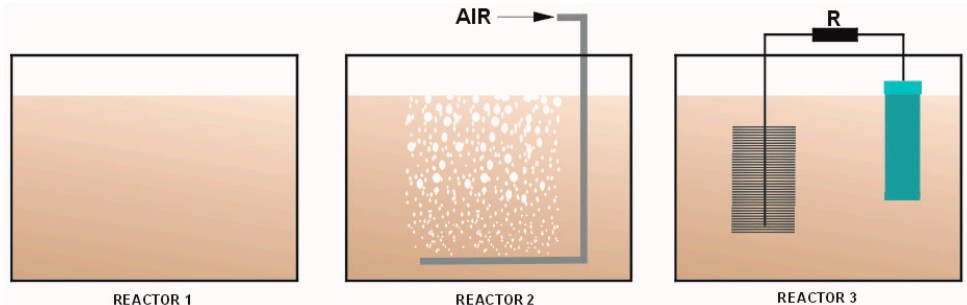

**Figure 16.** The three types of reactors used in the measurement of wastewater parameters during wastewater treatment: one excluding aeration (Reactor 1—R1), one with aeration (Reactor 2—R2), and as an MFC (Reactor 3—R3).

All reactors had the same dimensions (length/width/height: 40 cm × 20 cm × 20 cm). Thus, each reactor contained an equal volume of wastewater of 15 L. Each reactor worked separately but at the same time. The measurements of COD reduction were carried out (in each reactor, Figure 16) to a point where a 90% decrease in the concentration was obtained (Figure 10) [44,50], while the measurements of $NH_4^+$ and $NO_3^-$ concentrations were carried out over the same time as the shortest COD reduction time (15 days; reduction time for R1 reactor with aeration, Figure 10). Determining the measurement time for $NH_4^+$ and $NO_3^-$ parameters allowed a comparison of the results for all three reactors (Figures 11 and 12). In the reactor excluding aeration (R1), an interface between the wastewater and air occurred only at the wastewater surface. In the reactor with aeration (R2), aeration of wastewater was achieved as a result of using a pump with a capacity of 270 L·h$^{-1}$. In the last reactor design (R3), the treatment of wastewater occurred as a result of using an MFC. The wastewater parameters and electrical parameters were measured during the MFC operation. Figure 17 shows a schematic view of the MFC (Reactor 3).

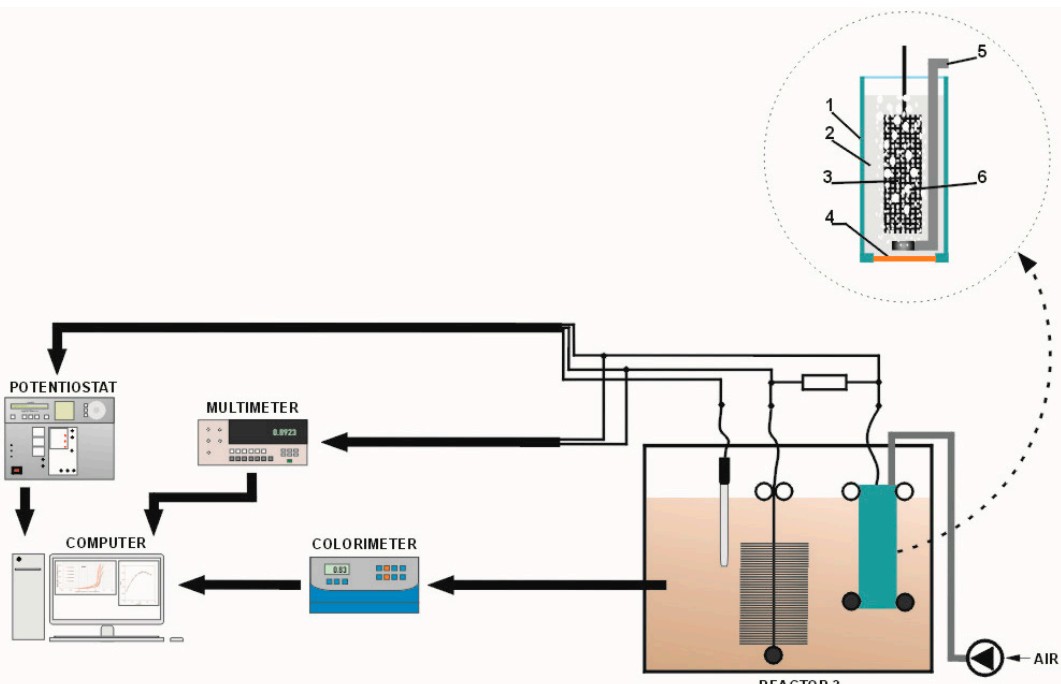

**Figure 17.** Schematic view of the MFC (Reactor 3, Figure 16): 1, casing; 2, electrolyte; 3, Cu–B cathode or the carbon cloth electrode; 4, proton exchange membrane (PEM); 5, air supply; 6, air bubbles.

Carbon cloth was used in the MFC in the anode, and the metal mesh with Cu–B catalyst formed the material applied in the design of the cathode of the system. For comparison, measurements were also carried out for a carbon cloth cathode. The surface area of the anode was 20 cm$^2$, while it was 15 cm$^2$

for the cathode. The electrical circuit of the MFC was constantly connected with a 10 Ω resistor [44,51]. The acclimation time of the microorganisms was 5 days [2,14,50].

The cathode was placed in a casing that was printed using 3D technology. The thickness of a single print layer was 0.09 mm. ABS (acrylonitrile butadiene styrene) filament was used as the material for 3D printing. The bottom wall was printed as a perforated wall in order to install a proton exchange membrane (PEM). After the PEM was installed, a catholyte with aqueous KOH solution (0.1 N) was filled into the casing. Subsequently, the cathode with Cu–B catalyst (or a carbon cloth cathode) was applied as the catholyte. Throughout the course of the experiment employing MFC (R3), the cathode was aerated at a capacity of 10 L·h$^{-1}$.

Nafion PF 117 (The Chemours Company, Wilmington, DE, USA), 183 μm thick, was used as the PEM. A Zortrax M200 printer (Zortrax S.A, Olsztyn, Poland) with Z-Suite software (Zortrax S.A, Olsztyn, Poland) was used to print the casing. A Hanna HI 83224 colorimeter (HANNA Instruments, Woonsocket, RI, USA) was applied for the measurement of wastewater parameters. An AMEL System 500 potentiostat (Amel S.l.r., Milano, Italy) with CorrWare software (Scribner Associates Inc, Southern Pines, NC, USA) and a Fluke 8840A multimeter (Fluke Corporation, Everett, WA, USA) were applied for the electrical measurements.

## 4. Conclusions

As demonstrated by the measurements, copper boride alloy is most suitable when oxidized for 6 h at a temperature of 673 K and when the concentration of boride is equal to 9%. In the MFC with a Cu–B cathode (9% of B), a current density of 0.35 mA·cm$^{-2}$ and power level of 6.11 mW were obtained. A shorter (by two days) COD reduction time (assumed reduction level: 90%) was shown in the MFC with a Cu–B cathode than in the MFC with a carbon cloth cathode. Moreover, the effectiveness of $NO_3^-$ reduction (during a 15 day period) in the MFC with a Cu–B cathode was higher than that in the MFC with a carbon cloth cathode. Thus, the higher catalytic activity of the Cu–B catalyst compared to carbon cloth and the possibility of using Cu–B alloy as a cathode catalyst in MFCs powered with municipal wastewater were shown in this paper.

**Author Contributions:** Data curation, P.P.W. and B.W.; investigation, P.P.W. and B.W.; methodology, P.P.W.; writing—review and editing, P.P.W. and B.W.; supervision, P.P.W. and B.W.

**Funding:** This research received no external funding.

**Conflicts of Interest:** The authors declare no conflict of interest.

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
