# Peer review of "Wastewater Treatment and Electricity Production in a Microbial Fuel Cell with Cu–B Alloy as the Cathode Catalyst"

_catalysts, doi:10.3390/catal9070572_

Round 1

Reviewer 1 Report

This is an interesting work on the use of a novel cathode catalyst in microbial fuel cells for wastewater treatment and enrtgy production. In my opinion this manuscript can be accepted in the present form

Author Response

The authors would like to thank the Reviewer for their valuable insights which helped the authors to improve this paper. In addition, language was corrected.

Reviewer 2 Report

I think that this work is an interesting work and it contributes to better understanding to wastewater treatment and electricity production in the microbial fuel  exploit the effect of using cathode based on combined Cu-B alloy on the microbial fuel cell activity. The work seems to have been well planned and performed and, in that sense, the manuscript is suitable for publication in catalysts after some changes for improving understanding and comprehension:

I suggest the authors to add more detailed discussion about methods section. Also more explanation about the bioelectrochemical process; experimental setup used for each microbial cell, hydraulic retention time, initial sludge characteristic and acclimation o microorganism etc.

“The speed of the process depends on the used of catalyst. In MFC a catalyst of anode are microbes (on carbon electrode), Thus, it is important to find catalyst for cathode. Due to its excellent catalytic properties platinum is most commonly used as the catalyst. However, due to the high price of platinum we should look, for other catalysts, as the non-precious metals”. The authors must include more reference in this part. I suggest to author:

In previous works (Liu et al., 2010, Yan et al, 2010, Erbay et al., 2015, Zuo et al., 2015, Raschintor et al., 2015 Asensio et al., 2017, Penteado et al., 2017, and Prestigiacomo et al.,, 2016) were compared the performance of microbial fuel cells (MFCs) equipped with different cheap electrode materials (graphite, carbon felt, foam, cloth and carbon nanotube sponges and  Polypyrrole/carbon blanck composite) during two-month long tests, in which they were operated under the same operating conditions. Despite using sp2 carbon materials (carbon felt, foam, and cloth) as anode in the different MFCs, results demonstrate that there are important differences in the performance, pointing out the relevance of the surface area and other physical characteristics on the efficiency of MECs. Differences It found not only in the production of electricity but also in the consumption of fuel. Carbon felt was found to be the most efficient anode material whereas the worst results were obtained with carbon cloth. Performance seems to be in direct relationship with the specific area of the anode materials. In comparing the performance of the MFC equipped with carbon felt and stainless steel as cathodes, the later shows the worst performance, which clearly indicates how the cathodic process may become the bottleneck of the MFC performance.

Discuss the advantages of the new cathode (Cu-B alloy) with this cheap carbon materials for use in microbial electrolysis cell.

On the other hand, recently, a number of studies have shown that ferrocyanide/ferricyanide is excellent cathodic electron acceptor, due to its good performance in the MFC (Penteado et al., 2017).  Discuss how the use of mediators would affect the stability of new cathode (Cu-B alloy).

The authors should use some references to clarify this statement

Zuo, K.,  Liang, S., Liang, P., Zhou, X., Sun, D., Zhang, X.,  Huang, X. Carbon filtration cathode in microbial fuel cell to enhance wastewater treatment (2015) BIORESOURCE TECHNOLOGY 185, 426-430

Y. Asensio, I.B. Montes, C.M. Fernández-Marchante, J. Lobato, P. Cañizares, M.A. Rodrigo. Selection of cheap electrodes for two-compartment microbial fuel cells (2017). JOURNAL OF ELECTRO ANALYTICAL CHEMISTRY 785, 235-240

C. Erbay, G. Yang, P. de Figueiredo, R. Sadr, C. Yu, A. Han, Three-dimensional porous carbon nanotube sponges for high-performance anodes of microbial fuel cells, Journal of Power Sources 298 (2015) 177-183.

Y. Yuan, S. Zhou, L. Zhuang, Polypyrrole/carbon black composite as a novel oxygen reduction catalyst for microbial fuel cells, Journal of Power Sources 195(11) (2010) 3490-3493.

E.D. Penteado, C.M. Fernández-Marchante, M. Zaiat, P. Cañizares, E.R. Gonzalez, M.A. Rodrigo(2017). Influence of carbon electrode material on energy recovery from winery wastewater using a dual-chamber microbial fuel cell. ENVIRONMENTAL TECHNOLOGY, 38, 1333-1341.

Y. Liu, F. Harnisch, K. Fricke, U. Schroeder, V. Climent, J. Miguel Feliu, The study of electrochemically active microbial biofilms on different carbon-based anode materials in microbial fuel cells, Biosensors & Bioelectronics 25(9) (2010) 2167-2171.

Author Response

The authors would like to thank the Reviewer for their valuable insights which helped the authors to improve this paper. 

After taking into account the suggestions of the Reviewer in the present work, the following scope of work was realized:

- language was corrected,

- the abstract was corrected,

- the introduction was expanded and corrected, 

- the introduction was supplemented with a wider description of the types of electrodes,

- the methodology followed in the study was corrected,

- the acclimation time was added, 

- discussion of the results was  reorganized and completed,

- the literature was supplemented.

In addition, these comments will also be taken into account when manuscripts are compiled in the future. 

Reviewer 3 Report

to search  In general the paper is well written and relevant references to the previous works in the field are well documented. In order to be suitable for the publication in Catalysts.

In line 15 the abbreviation COD should be define.

Author Response

The authors would like to thank the Reviewer for their valuable insights which helped the authors to improve this paper.

The abbreviation COD was defined. In addition, language was corrected.

In addition, these comments will also be taken into account when manuscripts are compiled in the future.